# Polydopamine Nanocluster Embedded Nanofibrous Membrane via Blow Spinning for Separation of Oil/Water Emulsions

**DOI:** 10.3390/molecules26113258

**Published:** 2021-05-28

**Authors:** Zhenglian Liu, Ziling Xu, Chaoqi Liu, Yajing Zhao, Qingyin Xia, Minghao Fang, Xin Min, Zhaohui Huang, Yan’gai Liu, Xiaowen Wu

**Affiliations:** 1Beijing Key Laboratory of Materials Utilization of Nonmetallic Minerals and Solid Wastes, National Laboratory of Mineral Materials, School of Materials Science and Technology, China University of Geosciences, Beijing 100083, China; liuzhenglian@outlook.com (Z.L.); Xzl98cugb@163.com (Z.X.); liuchaoqi0050@163.com (C.L.); zhaoyajing3166@163.com (Y.Z.); huang118@cugb.edu.cn (Z.H.); liuyang@cugb.edu.cn (Y.L.); xwwu@cugb.edu.cn (X.W.); 2School of Earth Science and Resources, China University of Geosciences, Beijing 100083, China; harry199529@hotmail.com

**Keywords:** nanofiber membrane, polydopamine nanocluster, blow spinning, oil–water emulsion

## Abstract

Developing a porous separation membrane that can efficiently separate oil–water emulsions still represents a challenge. In this study, nanofiber membranes with polydopamine clusters polymerized and embedded on the surface were successfully constructed using a solution blow-spinning process. The hierarchical surface structure enhanced the selective wettability, superhydrophilicity in air (≈0°), and underwater oleophobicity (≈160.2°) of the membrane. This membrane can effectively separate oil–water emulsions, achieving an excellent permeation flux (1552 Lm^−2^ h^−1^) and high separation efficiency (~99.86%) while operating only under the force of gravity. When the external driving pressure was increased to 20 kPa, the separation efficiency hardly changed (99.81%). However, the permeation flux significantly increased to 5894 Lm^−2^ h^−1^. These results show that the as-prepared polydopamine nanocluster-embedded nanofiber membrane has an excellent potential for oily wastewater treatment applications.

## 1. Introduction

Industrial development results in huge quantities of oily wastewater, causing water pollution and serious environmental problems and hindering sustainable development [1,2,3]. Oily wastewater typically originates from oil leakage, sewage discharges from industrial activities, and petroleum oil processes such as extraction, transportation, and processing [4,5,6]. Oily wastewater can be classified into slick, dispersed, emulsified, and dissolved oils [7,8]. Slick and dispersed oils can be effectively separated by static sedimentation and other methods with minimal processing difficulty due to their large oil droplet size. However, emulsified and dissolved oils form a stable dispersion system in water due to their extremely small particle size [9]. Furthermore, as the emulsified oil content is generally considerably higher than the dissolved oil content, the separation of emulsified oil is more crucial and urgent. Therefore, the effective separation of emulsified oil has become an important topic in oily wastewater treatment research.

Typical oily wastewater treatment methods include gravity sedimentation, centrifugation [10], air flotation, chemical flocculation, biodegradation, and adsorption [11]. These methods have several shortcomings, such as low separation efficiency, high treatment cost, high possibility of causing secondary pollution, and difficulty in treating oily wastewater containing high content of emulsions. On the other hand, using membrane separation for oily wastewater treatment can improves the defects mentioned above, which makes it an ideal method for oily-emulsion processing. The membranes currently used in this method can be classified into microfiltration, ultrafiltration, nanofiltration, and reverse-osmosis membranes [12]. However, the low flux and high working pressure still limit the practical use of separation membranes in oily wastewater treatment applications. Therefore, a filtration membrane with efficient separation performance is still required to overcome these shortcomings.

Recently, studies have been focusing on using nanofiber membranes for filtration and separation due to their high porosity and specific surface area [13,14,15,16,17,18]. Several studies have succeeded in the design and development of fibrous biomimetic materials with a superwetting surface [19,20,21]. Nanofiber membranes with hierarchical structure surfaces and high porosity offer numerous advantages, such as great super wettability, ability to separate small oil droplets, and high permeation flux.

According to Cassie’s model, when underwater–oil droplets are on a rough solid surface, water would penetrate the rough structure of the solid surface to form an interfacial layer of water–solid composite [22,23]. The water phase in the composite interface can repel the oil droplets to reduce the contact area between them and the solid surface. The three-phase contact line is discontinuous in this state, and oil droplets can easily roll on the surface, indicating their low adhesion to underwater solid surface. The Cassie state for oil droplets in water can be described as follows: cos θ* = *f* cos θ − 1 + *f* (where θ* is the apparent contact angle; θ is the intrinsic contact angle; *f* is the fraction of the solid in contact with the liquid). A membrane material with superhydrophilic and underwater super-oleophobic properties can be developed by constructing a rough hierarchical structure in the Cassie state on the surface.

Common nanofiber-fabrication techniques include electrospinning and solution blowing spinning (SBS). SBS has recently been proposed due to its several advantages such as high yield, simplicity, safety, and adaptability in large-scale production. In addition, compared to electrospinning, SBS offers higher efficiency and does not require high voltage, which increases the safety of the process [24,25]. SBS combines the advantages of electrospinning and traditional melt-blown technology [26,27]. Compared to electrospinning fiber membranes, SBS fiber membranes generally show higher porosity [26,28]. This method involves two parallel concentric fluid streams, a polymer solution, and a gas flow around the solution [29]. Polydopamine (PDA) is a well-known bio-inspired polymer. It can be easily deposited on the surface of various organic or inorganic materials and has good stability. Its abundant hydrophilic groups provide super-hydrophilic properties [30].

In this study, a porous fiber membrane with super wettability was developed and constructed using the SBS process with PDA clusters embedded on the surface. The rough surface structure and high porosity would ensure high separation efficiency and flux at a low driving pressure. This would enable the industrial application of the proposed modified membrane materials for oily wastewater treatment.

## 2. Results and Discussion

### 2.1. Preparation and Microstructural Characterization

An efficient design of separation membranes should ensure that the membrane surface has a hierarchical structure and a submicron pore structure to achieve superhydrophilicity and high water permeation flux, respectively [31,32]. To obtain a hydrophilic surface, the membrane was partially hydrolyzed using sodium hydroxide solution. The hierarchical structure was then constructed on the surface using embedded PDA nanoclusters to achieve super hydrophilicity. Based on Cassie’s model [23], this constructed hierarchical rough structure can achieve super hydrophilicity [33]. The nanofiber membrane after embedding PDA (H-PDA) exhibited superhydrophilicity in air with water contact angle (WCA) ≈ 0° and superoleophobicity with underwater–oil contact angle (UWOCA) ≈160° (Figure 1a,b), which are crucial for oil–water emulsion separation. Figure 1c indicates that polyacrylonitrile-nanofiber membranes with randomly oriented nonwoven structures (PAN) were successfully prepared using the SBS process. Figure 1d and e shows the SEM image of H-PDA, in which the diameter of the fiber ranges from 160 nm to 270 nm. The SEM image also shows that the PDA clusters were uniformly and randomly decorated on the surface of the nanofiber membrane. The inset is a high-magnification SEM of a PDA cluster with a diameter of approximately 2 μm.

The porosity of samples was characterized by the liquid absorption method [34]. The original PAN membrane’s porosity is 96.8%. The hydrolyzed PAN-nanofiber membrane (4 h)’s porosity is 84.5%. The nanofiber membrane after embedding PDA (20 h)’s porosity is 81.31%. The porosity of the PAN membrane is much higher than the porosity of the PAN obtained by electrospinning [21]. The thickness of membranes were 120 ± 10 μm, 102 ±4 μm, and 96 ± 5 μm, respectively.

Figure 2a shows a SEM image of PAN-nanofiber membrane hydrolyzed for 4 h by sodium hydroxide solution. No significant change was observed in its microstructure after the treatment. The rough microstructure on the membrane surface was further investigated by estimating the roughness values (Ra). Figure 2b demonstrates the Ra values of the membranes calculated using a noncontact optical profilometry analysis method. The Ra value of the original PAN-nanofiber membrane was 4.049 μm and decreased to 3.691 μm after partial hydrolysis of the membrane. This is not conducive for the construction of a hierarchical structure on the membrane surface. Figure 2c shows the pure-water flux data, which indicate that hydrolysis was not sufficient to change the water flux. The pure-water flux of the PAN membrane with no external driving pressure was 2021 Lm^−2^ h^−1^. However, the pure-water flux of hydrolyzed PAN-nanofiber membrane (PAN (H)) decreased to 1608 Lm^−2^ h^−1^. This can be attributed to the shrinkage caused by the capillary tension during the drying process of the nanofiber membrane, which eventually causes a decrease in porosity. To ensure a hierarchical structure on the surface, another step was conducted to construct a rough surface. The SEM images in Figure 3a–d show the surface microstructure of the H-PDA membranes prepared at different polymerization times (1, 4, 8, and 20 h) in a dopamine hydrochloride solution (2 g/L) bath, respectively. As shown in the SEM image in Figure 3a, no significant difference was observed after conducting the treatment for 1 h. Nanoclusters started to occur and increase with time. After a polymerization for 20 h, PDA nanoclusters were randomly embedded on the surface of the nanofiber membrane. The Ra value of H-PDA (after polymerization for 20 h) reached 6.043 μm. These nanoclusters embedded on the surface of the nanofiber membrane slightly increased the pure-water flux from 1608 Lm^−2^ h^−1^ to 2210 Lm^−2^ h^−1^.

### 2.2. Optimization of Surface Wettability

A highly selective wettability for oil and water is a crucial factor for materials used for oil–water separation [35,36]. Thus, the wettability of the fabricated membranes was examined. Although the nitrile group in PAN is hydrophilic, the contact angle measurements indicated that it was still high (53°), which does not meet the requirements for efficient separation. The treatment with NaOH solution (1 mol/L) was conducted to partially hydrolyze the nitrile group in PAN into amide and carboxyl groups. Figure 3e shows the FT-IR spectrum of the PAN-nanofiber membrane treated at different times. The peak at 1732 cm^−1^ can be attributed to the stretching vibration of the carbonyl group, which proves a successful hydrolysis of APN. Further, the effect of the hydrolysis time was studied. Figure 3f shows the contact angle in the air plotted versus the hydrolysis time. The contact angle was reduced from its initial value of 53° to 30° after 4 h of hydrolysis. After hydrolysis for 2 h, smaller change was observed in the contact angle. After 4 h, a very slight reduction was observed in the contact angle. To further improve the hydrophilicity and underwater oleophobic properties, hydrophilic PDA nanoclusters were polymerized on the surface of PAN (H). The hierarchical rough surface constructed on the membrane further improved the wettability while increasing the underwater oleophobic properties. The abundance of phenolic hydroxyl groups in the PDA further reduced the WCA of the fiber membrane after the PDA treatment. Figure 4a shows the contact angle of PAN (H) membrane in air after dopamine polymerization for different times. Increasing the time to 20 h reduced the contact angle to 18.2°. Moreover, H-PDA exhibited a super wettability. Figure 4b shows the dynamic contact angle process of PAN, PAN (H), and H-PDA, and water droplets were able to rapidly permeate through H-PDA. The WCA reached nearly 0° only 0.25 s after the water droplet contacted the surface of H-PDA, which is a significantly shorter time than those of PAN and PAN (H). The digital photo images of the dynamic contact angle process of H-PDA (Figure 4e) indicate that the entire process took place within 0.25 s.

H-PDA exhibited underwater oleophobicity. The underwater–oil contact angle of the original PAN-nanofiber membrane was 147.83°. By extending the time of dopamine polymerization, the underwater–oil contact angle increased to 160.2° (Figure 4c). In Cassie’s equation, θ* and θ were substituted using the values of underwater–oil contact angles of H-PDA and PAN, respectively, to calculate *f*, which was estimated to be *f* ≈ 0.37. Thus, the contact area between the PDA-treated fiber membrane surface and the oil droplet was reduced.

The FT-IR spectrum (Figure 4c) indicates the successful synthesis of PDA. The intensity of the N–H stretching vibration at 3120–3680 cm^−1^ increased due to the introduction of PDA, which contains abundant –NH_2_ groups [21,37]. Figure 4f shows an adequate contact between an oil droplet (n-hexane) and the membrane surface. When the oil droplet encountered a significant deformation, the oil droplet slowly raised. During this lifting process, the oil droplet remained spherical in shape with no visible deformation, demonstrating low oil adhesion to H-PDA. To further demonstrate the anti-oil properties of the membrane, the soy oil (dyed by Sudan III) was quickly ejected onto the membrane under water, and the oil jet immediately bounced up from the surface without any adhesion (Figure 4g).

### 2.3. Evaluation of the Oil–Water Emulsion Separation Performance

Based on the previous results, the membranes exhibited superhydrophilicity and underwater oleophobicity, and hence, are capable of separating oil–water emulsions. Three types of emulsions were used to evaluate the performance of the membranes: n-hexane, diesel, and soy oil emulsions. Figure 5a shows the process of emulsion separation under an external driving pressure. The membranes were placed into the button of a cell, and then the cell was assembled. The membranes were prewetted, then the emulsions were poured while regulating the driving pressure to the target value. The filtrates were collected to further evaluate the separation performance. The separation under the force of gravity was evaluated by removing the pressure regulator and maintaining a constant height difference between the outlet of the cell and the end of the tube. Here, the height difference was maintained at 30 cm, which equals a driving pressure of approximately 3 kPa. The milky oil–water emulsions became transparent (Figure 5b), and no oil droplets were observed in the filtrates, illustrating that the oil droplets have been successfully removed from the oily wastewater by H-PDA. The permeation fluxes of n-hexane-, diesel-, and soy oil-based emulsions were 1443.6 Lm^−2^ h^−1^, 1645.0 Lm^−2^ h^−1^, and 1552.9 Lm^−2^ h^−1^, respectively (Figure 5c). Table 1 is a comparison of total organic carbon (TOC) content of three emulsions before and after filtration. The separation process was conducted using H-PDA under only the force of gravity. Further comparative experiments revealed that the permeation flux increased with the increase in the driving pressure, but the separation performance was hardly affected. By increasing the driving pressure to 20 kPa, the TOC content of the filtrate separated by PAN membrane sharply increased to 334.1 mg L^−1^ compared to 55.01 mg L^−1^ under gravity only (Figure 5d and Table 2), and the separation efficiency decreased from 98% to 91%. On the other hand, H-PDA showed better separation efficiency (up to 99.86% using gravity only). Moreover, by increasing the driving pressure to 20 kPa, the TOC content of the filtrate reached 7.77 mg L^−1^, which is a slight increase compared to the TOC (5.64 mg L^−1^) obtained under gravity only, and the separation efficiency remained above 99.8%. In addition, the permeation flux increased from 1552.9 Lm^−2^ h^−1^ to 5864.6 Lm^−2^ h^−1^, i.e., a 3.8-time increase. These separation fluxes were one order of magnitude higher than those of commercial microfiltration (MF) and ultrafiltration (UF) membranes and were superior to those of previously reported membranes with similar rejection properties and driven external pressures [38,39,40].

## 3. Materials and Methods

### 3.1. Materials

Polyacrylonitrile (PAN, Mw = 250,000) was obtained from Dow Chemical (Shanghai, China) Co., Ltd., while the tris-hydrochloride buffer (1M, pH = 8.5), dopamine hydrochloride, sodium hydroxide (NaOH), N, N-Dimethylacetamide (DMAC), n-hexane, Sudan Ⅲ were purchased from Aladdin Chemistry Co. Ltd., Shanghai, China. Diesel oil was provided by the China National Petroleum Corporation, and soy oil was obtained from a local supermarket. All chemicals were of analytical grade and used as received without further purification.

### 3.2. Fabrication of PAN-Nanofiber Membranes

Figure 6 shows a schematic illustration of the formation process of nanofiber membranes. The precursor solution was prepared by dissolving a sufficiently dried PAN powder in DMAC at a concentration of 13 wt% under stirring for 12 h at 60 °C. The solution was left to set for 10 h to eliminate air bubbles. Next, an SBS machine manufactured in our laboratory was used to fabricate nanofiber membranes. In SBS, the precursor solution was loaded into a syringe capped with 30G metal-dispensing needle operated at a fixed rate of 1.5 mL/h. The needle tip was coaxially fixed with the air nozzle (diameter = 2 mm). The spinning air flow was supplied by an air compressor equipped with an adsorption dryer. A flowmeter was used to ensure a stable and sufficient spinning airflow at a rate of 0.6 m³/h. The syringe reciprocated horizontally at a frequency of 0.5 Hz with a reciprocating range of 15 cm. The fiber was collected by polypropylene nonwoven fabrics coated on a porous, metallic rotating roller (rotation rate = 200 rpm). The porous roller was connected to a centrifugal fan to create suction and form a negative air pressure. Further, the porous roller was grounded. To provide positive charge to the fibers and ensure attraction between the fibers and grounded roller, a 5-kV voltage was applied to the tip of the needle for efficient fiber collection. The spinning process was conducted for 3 h, after which the obtained membrane was dried at 60 °C for 2 h. The size of original PAN membrane is 29 cm long and 20 cm wide.

### 3.3. Construction of Superhydrophilic Nanofiber Membranes

The dried membranes constructed on polypropylene nonwoven fabrics substrate were treated in a NaOH solution (500 mL, 1 mol/L) at 20 °C for different times. The obtained membranes were then rinsed using deionized water until their pH reached 7 [19,29]. Next, they were placed on the button of the container, where a dopamine hydrochloride solution (prepared using tris-hydrochloride buffer (10 mM) as the solvent) (500 mL, 2 g/L) was slowly poured [40,41]. The membranes were left submerged in the dopamine hydrochloride solution for different times. The membranes were again rinsed using deionized water and dried at 40 °C for 2 h.

### 3.4. Preparation of Emulsions

The oil–water emulsions were prepared by mixing oils and water under vigorous stirring at a speed of approximately 2000 rpm, a mass ratio of 1:99, and a temperature range of 18–22 °C for 1 h until emulsified, milky solutions were obtained. The obtained emulsions were stable for more than 20 h. The actual TOC content of each emulsion before filtration is tested to calculate the filtration efficiency afterwards.

### 3.5. Oil–Water Emulsion Separation Experiments

The emulsion separation performance of the membranes was evaluated using an Amicon stirred cell (UFSC05001, Merck Millipore) with a pressure regulator connected to a compressed-air supply. The membranes were fixed into the cell so that the effective separation area was 10.18 cm^2^. A polyethylene (PE) tube (inner diameter = 1 mm and length = 30 cm) was connected to the outlet of the cell and was laid down vertically. A beaker was placed under the end of the tube to collect the filtrate. Finally, the total organic carbon (TOC) content of the feed solutions and the corresponding filtrates was measured to evaluate the separation efficiency.

### 3.6. Characterizations

The microstructures of the membranes were characterized using scanning electron microscopy (SEM, SUPRA55, ZEISS, Oberkochen, Germany). For chemical bond characterization, Fourier transform infrared (FT-IR) spectroscopy was also performed (Spectrum One, PerkinElmer, Boston, MA, USA). The topographic roughness parameters (Ra) were measured using a noncontact optical profilometry (Countor GT K, Bruker, Billerica, MA, USA). The water contact angle (WCA) and underwater–oil contact angle (UWOCA) were measured using a contact angle goniometer (JC2000D, Shanghai Zhongchen, China). During the testing process, the test liquid (2 μL) was dispensed at the membrane surface. The oil–water emulsions were characterized by optical microscopy (PSM-1000, Motic, Wetzlar, Germany). The TOC content in the feed solutions and the corresponding filtrates was measured using a TOC analyzer (TOC-L CPH/CPN, Shimadzu, Kyoto, Japan).

## 4. Conclusions

In summary, the PAN-nanofiber membranes were successfully prepared using a solution blowing spinning (SBS) process. A superhydrophilic and underwater super-oleophobic hierarchical surface structure was obtained by further hydrolysis treatment and polymerization of dopamine on the membrane surface. The Polydopamine (PDA)-nanocluster successfully enhanced the hydrophilicity of the nanofiber membrane. This PDA-nanocluster-embedded membrane can effectively separate various oil–water emulsions. The proposed membranes exhibited high separation efficiency, ultrahigh permeation flux, and good preferable antifouling performance. Combining these benefits with the potential of the SBS method for large-scale manufacturing makes the proposed strategy a convenient and powerful tool to fabricate superhydrophilic fibrous materials for practical applications.

## Figures and Tables

**Figure 1 molecules-26-03258-f001:**
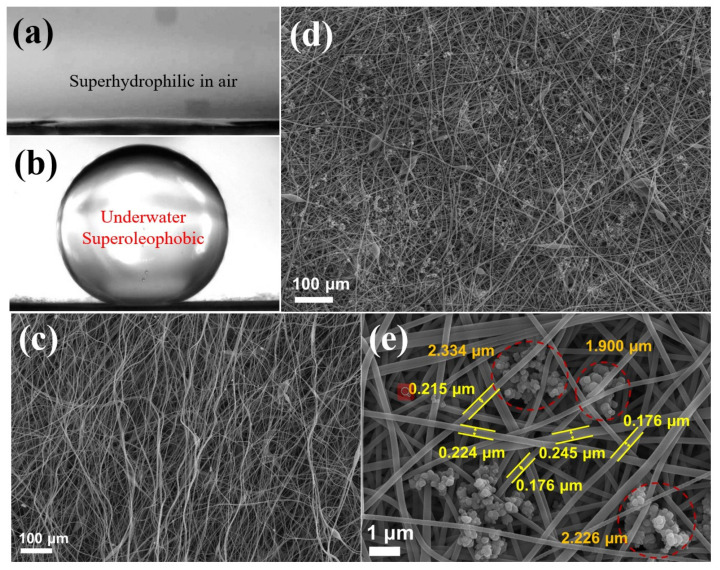
(**a**,**b**) Illustrations of a water droplet in air and another under water on the surface of H-PDA. (**c**) SEM image of the original PAN nanofiber formed using an SBS process. (**d**) SEM image of the surface of H-PDA. (**e**) Enlarged SEM image of the surface embedded with PDA nanoclusters.

**Figure 2 molecules-26-03258-f002:**
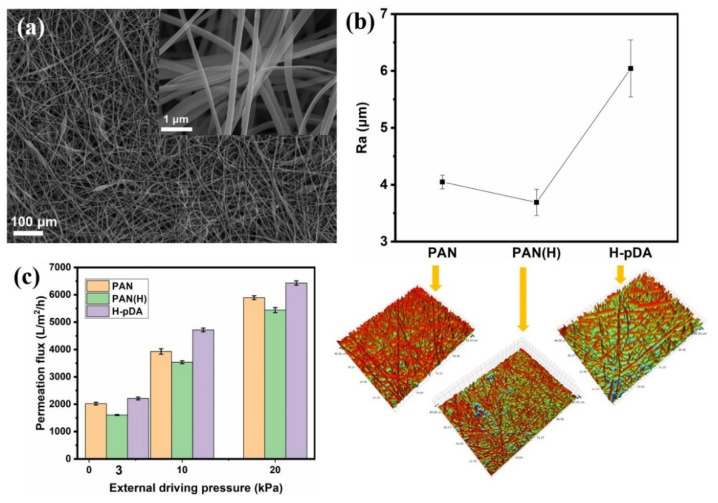
(**a**) SEM image of the PAN membrane treated with sodium hydroxide solution for 4 h. The inset is a high-magnification SEM image of the fiber surface. (**b**) Ra values of the membranes (PAN, PAN after hydrolyzation for 4h, and hydrolyzed PAN after 20 h of PDA treatment.). There are noncontact optical profiler imagers, respectively, underneath the graph. (**c**) Pure-water flux of PAN, PAN-H, and H-PDA at different external driving pressure.

**Figure 3 molecules-26-03258-f003:**
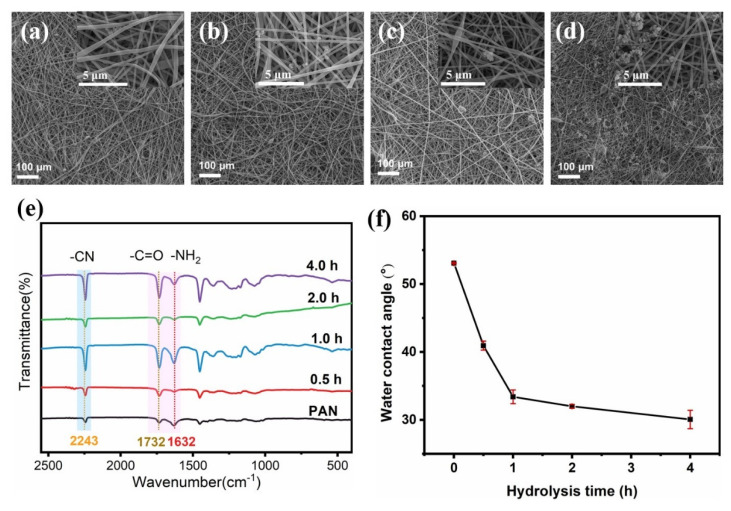
(**a**–**d**) SEM images of the hydrolyzed PAN with PDA on the surface after polymerization for different times (1, 4, 8, and 20 h, respectively). (**e**) FT-IR spectrum of the nanofiber membranes after different hydrolysis times. (**f**) Water contact angle of the nanofiber membranes prepared at different hydrolysis times.

**Figure 4 molecules-26-03258-f004:**
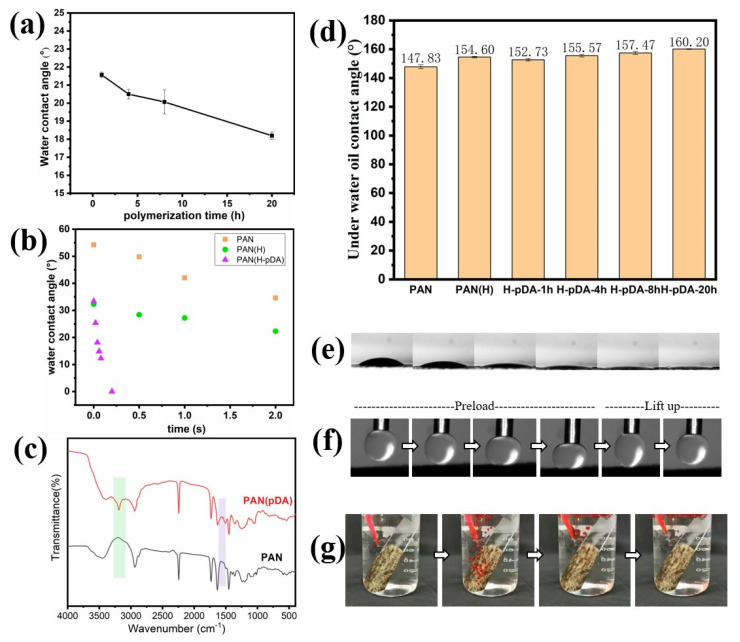
(**a**) Contact angles measured in air of H-PDA prepared at different polymerization times. (**b**) Water contact angles of different membranes versus drop time. (**c**) FT-IR spectrum of PAN and H-PDA. (**d**) Under water oil contact angle of different membranes. (**e**) Digital photos of the changes in the water contact angle changes after the drop of the droplet. The entire process shown by the images occurred within 0.25 s. (**f**) Dynamic underwater–oil adhesion of H-PDA. (**g**) Real-time images recording the superior anti-oil fouling performance of H-PDA. Soy oil is the oil used in the panels in (**g**).

**Figure 5 molecules-26-03258-f005:**
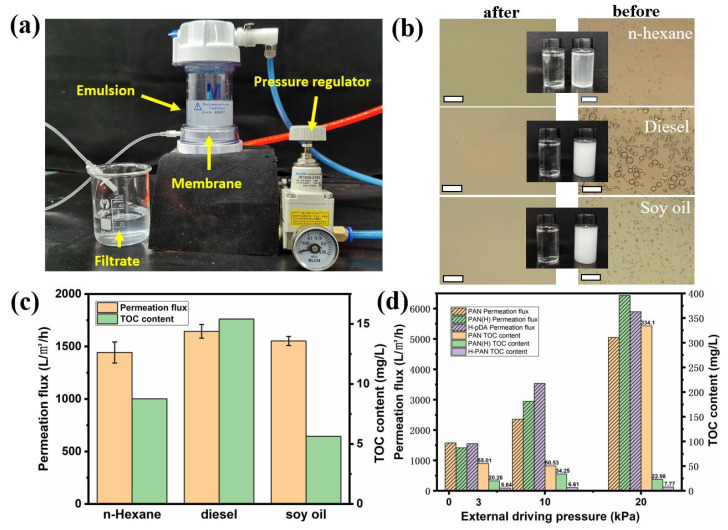
(**a**) Digital photos showing the emulsion separation process. (**b**) Optical microscopy images and photographs of the emulsions before and after separation (the scale bar is 50 μm). (**c**) Permeation flux and the corresponding TOC content of the filtrates for various emulsions under the effect of gravity (no external driving force). (**d**) Permeation flux and the corresponding TOC contents of the filtrates from the separation process of soy oil emulsion using different membranes at different driving pressures.

**Figure 6 molecules-26-03258-f006:**
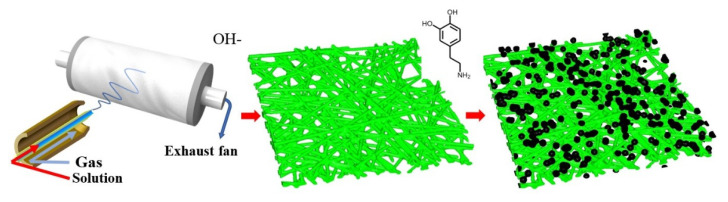
Schematic illustration of the formation process of the nanofiber membranes.

**Table 1 molecules-26-03258-t001:** The TOC content of the filtrates for various emulsions under the effect of gravity (no external driving force) using H-PDA.

Emulsion	TOC of before Filtration (mg/L)	TOC of after Filtration (mg/L)
n-Hexane	207.4 ± 24.9	8.77 ± 0.08
diesel	2288 ± 27.23	15.42 ± 0.31
soy oil	4132 ± 29.56	5.64 ± 0.46

**Table 2 molecules-26-03258-t002:** The TOC contents of the filtrates from the separation process of soy oil emulsion using different membranes at different driving pressure.

Membrane	TOC of before Filtration (mg/L)	TOC of after Filtration (mg/L)		
		Gravity	10 kPa	20 kPa
PAN	4132 ± 29.56	55.01 ± 0.76	50.53 ± 0.97	334.1 ± 0.07
PAN(H)		20.26 ± 0.49	34.25 ± 0.11	22.98 ± 0.37
H-PAN		5.64 ± 0.46	6.61 ± 0.27	7.77 ± 0.73

## Data Availability

Not applicable.

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
