# Peer review of "Polydopamine Nanocluster Embedded Nanofibrous Membrane via Blow Spinning for Separation of Oil/Water Emulsions"

_molecules, 2021, doi:10.3390/molecules26113258_

Round 1
Reviewer 1 Report
Liu et al. prepared a polydopamine nanocluster embedded nanofibrous membrane via blow spinning and tested it for separation of oil/water emulsions.
This work certainly contains some valuable inputs in the context of separation of oil/water emulsions by suitable membranes. However, in my opinion the manuscript is not well written and it must be improved. The authors must revise it under careful considerations in order to make this scientific work publishable, with support of some suggested comments and questions.
- the manuscript is very difficult to read, it is described in a confused manner and the reader struggles to read and understand: it is not fluid especially in some parts. So, it must be improved in the presentation and in sequential logic of discussion. All Figures includes subfigures (i.e. from A to E, etc.) but in the text the order of mentioning them is altered (B, D, C, A …), and then the first figure, 1A, is mentioned last (in paragraph 3). So, the authors should properly arrange the consequent order of figures and subfigures in the text (i.e., Figure 1 a, 1b, 1c ….; Figure 2a, 2b, 2c.., and so on) or, alternatively, the distribution of subfigures in the Figures. The text must be consequently better formulated.
- the driving force of this work should be the possibility to prepare highly efficient porous membrane for oil/water separation by using SBS technique, because the use of PDA nanocluster is not a novelty in this context (i.e., ref 21) However, in my opinion, this point is not well enhanced and discussed (in results and discussion this step is completely ignored).
- the “Materials and methods” paragraph would be preferable to put it before “Results and Discussions”. However, if this is not possible due to editorial guidelines, please define acronyms the first time they appear (i.e., TOC, PAN, etc.) in results and discussion.
- lines 39-45: shortcomings and advantages are listed at the same modality, so it seems quite repetitive. Please, improve this period.
- The effect of PDA nanocluster is discussed in lines 152-154 in terms of improved wettability due to hydroxyl groups. why couldn't another polymer with similar functional groups do? The authors should better discuss the reasons behind this choice; PDA is a well-known bio-inspired polymer with suitable properties, first of all adhesion and simple preparation method. Specific references on PDA are missing, so at least one could be added and correctly linked to discussion (for this see also next point).
- The H-PDA membranes were prepared at different polymerization times (1, 4, 8, and 20 h) in a dopamine hydrochloride solution (2 g/L) bath, respectively. Please clarify how and why you have selected these reaction conditions. Furthermore, could longer reaction times of basic hydrolysis furtherly improve hydrophilicity of membranes?
- References must be carefully checked:
In line 84, “…respectively [29] [31]” but ref 30 is not yet appeared.
Then, in relation to the text and position: several references are not appropriately correlated to the statements of the text. Only for example:
-line 173, ref 29: this reference does not deal with PDA nor does it report IR characterization of the polymer
- lines 256-257, ref 30: this reference this reference does not deal with PDA nor does it report it experimental procedure
- Some pictures are not very clear:
SEM images in Figures 1 and 3 should be enlarged (as well done in Figure 2) in order to appreciate the features of fibres described in the text. Then, in the text the authors said that “Figure 1c shows the SEM image of H-PDA, in which the average diameter of the fiber was 300 nm”, but in the figure this is not observable. How did you measure it? Then, no fiber size was reported for PAN sample: a comparison can be appropriate.
Moreover, in Figure 5a the red characters are not well visible in contrast of dark picture: I could suggest to use yellow used for arrows for example.
- In manuscript, membranes are defined as porous. Have the authors data on the porosity of sample (i.e., pores size and/or their distribution) and thickness of membrane?
- The authors declare that different reaction conditions for DA polymerization were explored to optimize the synthesis, and they give only some generic indications (without specific data) on the effect of DA conc., temperature, buffer pH. In my opinion, complete data (of different experimental conditions performed) and results should be reported and compared to properly support assessment and draw conclusions.
- Some important information in materials and methods should be reported: size of membrane samples obtained and then used for further steps, volume of used solution (i.e., NaOH, Tris solution of DA).
- In order to validate the effectiveness of the membrane in the oil/water separation, test on surfactant-stabilized emulsions (more stable and difficult to separate) should be performed. Do the authors have any data on this? Furthermore, what about the recyclability of prepared membranes?
- The concept of antifouling performance is only mentioned in conclusions, so it should be clarified and a little discussed before.
- The manuscript shows many mistakes occurring in the text, so it needs to be carefully checked in terms of: spaces, punctuation, reference and measure unit formatting in overall text: i.e., superscript characters in measure units (Lm-2h-1, etc.) or subscript one in chemical formula (-NH2).
- The part at the end of manuscript (author contributions, … sample availability) must be completed.
Author Response
Response to Editor and Reviewers
Molecules
Manuscript ID: molecules-1201771
Manuscript Title: Polydopamine nanocluster embedded nanofibrous membrane via blow spinning for separation of oil/water emulsions
Dear Editor and Reviewers,
We are grateful for your kind help and constructive suggestions on our manuscript. We have studied the reviewer’s comments carefully and have made the corrections accordingly.
Starting from page 2 in this file, we provide the point-by-point replies to the reviewer’s comments. Meanwhile, the modified contents have been highlighted in the revised manuscript (in red) for your consideration. We hope our efforts on modification will make our work more appropriate for publication.
Thank you for your attention and consideration. We are looking forward to hearing from you.
Sincerely yours,
Minghao Fang
Response to the reviewer 1:
Authors reply:
We thank the reviewer for providing constructive feedback to improve our manuscript. We have gone through the reviewers’ comments carefully and have made the modifications accordingly. We highlighted the modified contents in red in the revised manuscript. Our point-by-point responses to the reviewer’s comments are provided here.
Comment 1:
the manuscript is very difficult to read, it is described in a confused manner and the reader struggles to read and understand: it is not fluid especially in some parts. So, it must be improved in the presentation and in sequential logic of discussion. All Figures includes subfigures (i.e. from A to E, etc.) but in the text the order of mentioning them is altered (B, D, C, A …), and then the first figure, 1A, is mentioned last (in paragraph 3). So, the authors should properly arrange the consequent order of figures and subfigures in the text (i.e., Figure 1 a, 1b, 1c ….; Figure 2a, 2b, 2c.., and so on) or, alternatively, the distribution of subfigures in the Figures. The text must be consequently better formulated.
Authors reply:
We are grateful for the reviewer’s comment. We have reorganized the order of the figures in the revised manuscript. Arranged the figures in order of discussion.
Comment 2:
the driving force of this work should be the possibility to prepare highly efficient porous membrane for oil/water separation by using SBS technique, because the use of PDA nanocluster is not a novelty in this context (i.e., ref 21) However, in my opinion, this point is not well enhanced and discussed (in results and discussion this step is completely ignored).
Authors reply:
We thank the reviewer for the useful comment. Inspired by ref 20, the rough surface is helpful to further improve the hydrophilicity. In our work, polydopamine is synthesized as clusters embedded on the surface of the fiber membrane to improve the surface roughness and achieve better hydrophilicity which is slightly different from the way in ref 21 that PDA wase polymerized on the surface of each single nanofiber.
The advantages of SBS technology do need to be enhanced and discussed. In addition to high production efficiency, relatively higher porosity is also an important advantage of SBS, because in emulsion separation applications, high porosity is a guarantee of high permeation flux. The fiber membrane in our work shows higher porosity and flux compared with the membrane in ref 21. Related modifications are also added in the revised manuscript in “introduction” and “result and discussion.”
Comment 3:
the “Materials and methods” paragraph would be preferable to put it before “Results and Discussions”. However, if this is not possible due to editorial guidelines, please define acronyms the first time they appear (i.e., TOC, PAN, etc.) in results and discussion.
Authors reply:
We are grateful for pointing it out. The editorial guidelines require these contents to be placed behind of the “Results and Discussions”. We have defined acronyms the first time they appear (WCA at line 90, UWOCA at line 91, TOC at line 211).
Comment 4:
lines 39-45: shortcomings and advantages are listed at the same modality, so it seems quite repetitive. Please, improve this period.
Authors reply:
We appreciate the reviewer’s suggestion. We have reorganized the descriptions (in line 42-43).
Comment 5:
The effect of PDA nanocluster is discussed in lines 152-154 in terms of improved wettability due to hydroxyl groups. why couldn't another polymer with similar functional groups do? The authors should better discuss the reasons behind this choice; PDA is a well-known bio-inspired polymer with suitable properties, first of all adhesion and simple preparation method. Specific references on PDA are missing, so at least one could be added and correctly linked to discussion (for this see also next point).
Authors reply:
We thank the reviewer for the useful comment. We did choose to use PDA based on such considerations of all adhesion and simple preparation method, and added relevant expositions in the “introduction”, and cited articles in related fields to support our choice(ref 29).
Comment 6:
The H-PDA membranes were prepared at different polymerization times (1, 4, 8, and 20 h) in a dopamine hydrochloride solution (2 g/L) bath, respectively. Please clarify how and why you have selected these reaction conditions. Furthermore, could longer reaction times of basic hydrolysis furtherly improve hydrophilicity of membranes?
Authors reply:
We are grateful for the reviewer’s comment. After our research, polymerization time is the most important influencing factor, so we choose polymerization time as the core variable for research (ref 21 1 and ref 41). It is a more convenient treatment to improve the hydrophilicity by changing the polymerization time. And according to the results, continuing to extend the polymerization time has a very limited improvement in hydrophilicity. According to the results of Figure 4, the polymerization time is changed from 8 hours to 20 hours, and the change trend of the decrease in contact angle is much slower.
Comment 7, 8 & 9:
References must be carefully checked:
In line 84, “…respectively [29] [31]” but ref 30 is not yet appeared.
Then, in relation to the text and position: several references are not appropriately correlated to the statements of the text. Only for example:
-line 173, ref 29: this reference does not deal with PDA nor does it report IR characterization of the polymer
lines 256-257, ref 30: this reference this reference does not deal with PDA nor does it report it experimental procedure
Authors reply:
We appreciate the reviewer’s comment. We have corrected these mistakes in the revised manuscript. And a new reference is added (ref 41).
Comment 10:
Some pictures are not very clear:
SEM images in Figures 1 and 3 should be enlarged (as well done in Figure 2) in order to appreciate the features of fibres described in the text. Then, in the text the authors said that “Figure 1c shows the SEM image of H-PDA, in which the average diameter of the fiber was 300 nm”, but in the figure this is not observable. How did you measure it? Then, no fiber size was reported for PAN sample: a comparison can be appropriate.
Moreover, in Figure 5a the red characters are not well visible in contrast of dark picture: I could suggest to use yellow used for arrows for example.
Authors reply:
We are grateful for the reviewer’s comment. We have reorganized the order of the figures in the revised manuscript. We have also improved the display of these figures and corrected some descriptions. Some typical size information is also added to the figures.
Comment 11:
In manuscript, membranes are defined as porous. Have the authors data on the porosity of sample (i.e., pores size and/or their distribution) and thickness of membrane?
Authors reply:
We appreciate the reviewer’s comment. We test the porosity of samples using the liquid absorption method. The test method is detailed in ref 33. The original PAN membrane’s porosity is 96.8%. The hydrolyzed PAN-nanofiber membrane (4 hours)’s porosity is 84.5%. The nanofiber membrane after embedding PDA (20 hours)’s porosity is 81.31%. the thickness of membranes were 120 ±10 μm, 102 ±4 μm and 96 ±5 μm respectively.
Comment 12:
The authors declare that different reaction conditions for DA polymerization were explored to optimize the synthesis, and they give only some generic indications (without specific data) on the effect of DA conc., temperature, buffer pH. In my opinion, complete data (of different experimental conditions performed) and results should be reported and compared to properly support assessment and draw conclusions.
Authors reply:
We appreciate the reviewer’s comment. This study only focused on the polymerization time as a research variable, and other parameters were not the focus of the study. The concentration of DA, temperature and pH of the buffer are all based on previous research results, not as the research goals of this article. The core goal is to control the hydrophilicity through the change of polymerization time, so this goal can be achieved through the change of polymerization time only.
Comment 13:
Some important information in materials and methods should be reported: size of membrane samples obtained and then used for further steps, volume of used solution (i.e., NaOH, Tris solution of DA).
Authors reply:
We thank the reviewer for the comment. We measured the size of membrane and added relevant data. the size of original PAN membrane is 29 cm long and 20cm wide (Determined by the equipment performance). Each of them was add to the manuscript in corresponding position.
Comment 14:
In order to validate the effectiveness of the membrane in the oil/water separation, test on surfactant-stabilized emulsions (more stable and difficult to separate) should be performed. Do the authors have any data on this? Furthermore, what about the recyclability of prepared membranes?
Authors reply:
We thank the reviewer for the useful comments. This is indeed a very important research point. Regrettably, this article did not make further research on this aspect. We plan to focus on this aspect in further research.
Comment 15:
The concept of antifouling performance is only mentioned in conclusions, so it should be clarified and a little discussed before.
Authors reply:
We appreciate the reviewer’s comment. The antifouling performance of the membrane is mainly reflected in the oil repellent performance and low oil adhesion, which is discussed in section 2.2 and Figure 4.
Comment 16:
The manuscript shows many mistakes occurring in the text, so it needs to be carefully checked in terms of: spaces, punctuation, reference and measure unit formatting in overall text: i.e., superscript characters in measure units (Lm-2h-1, etc.) or subscript one in chemical formula (-NH2).
Authors reply:
We appreciate the reviewer’s comment. These may have occurred when the document was submitted, and we have corrected it seriously.
Comment 17:
- The part at the end of manuscript (author contributions, … sample availability) must be completed.
Authors reply:
We appreciate the reviewer’s comment. We have completed this part of the manuscript.

Reviewer 2 Report
The Authors presented an interesting idea of the membrane composition. First, is this manuscript belong to some special issue? If not, I will consider switching this manuscript for Processes, Applied Sciences, Materials, Membranes, or Nanomaterials. Second, there are a few details that should be explained before publication, and major revision is needed.
I have to pointed that additional tests of the porosity will strongly increase the manuscript quality. If it is not possible, please clearly state why did not you make this examination.
Table with emulsion content description is necessary. Additional details should be presented nor in the materials and methods section.
Did you use some surfactants/stabilizers for emulsion preparation?
Did you measure/obtain the porosity of the membrane?
Please explain what role plays here polydopamine?
Additional tests which will show the oil content (or other components) before and after filtration are necessary to prove the membrane efficiency.
Abstract - please avoid the words such as "outstanding, excellent, etc." you can use a high efficiency, etc.
It must be pointed, that I really enjoyed all figures - most of them are well organized. I m truly happy, that the Authors showed how the system works, as presented in fig. 5. Nevertheless, please add the values on the scale bars as well as error bars. Please verify fig. 5d where is before and what is after permeation?
Fig. 3f please enlarge the Ra images.
Uniform the colors on the graphs for each sample.
Author Response
Response to Editor and Reviewers
Molecules
Manuscript ID: molecules-1201771
Manuscript Title: Polydopamine nanocluster embedded nanofibrous membrane via blow spinning for separation of oil/water emulsions
Dear Editor and Reviewers,
We are grateful for your kind help and constructive suggestions on our manuscript. We have studied the reviewer’s comments carefully and have made the corrections accordingly.
Starting from page 2 in this file, we provide the point-by-point replies to the reviewer’s comments. Meanwhile, the modified contents have been highlighted in the revised manuscript (in red) for your consideration. We hope our efforts on modification will make our work more appropriate for publication.
Thank you for your attention and consideration. We are looking forward to hearing from you.
Sincerely yours,
Minghao Fang
Response to the reviewer 2:
The Authors presented an interesting idea of the membrane composition. First, is this manuscript belong to some special issue? If not, I will consider switching this manuscript for Processes, Applied Sciences, Materials, Membranes, or Nanomaterials. Second, there are a few details that should be explained before publication, and major revision is needed.
Authors reply:
We thank the reviewer for the useful comments. Based on the suggestions, we have revised our work carefully with corresponding corrections and additions. We highlighted the modified contents in the revised manuscript.
Comment 1&4:
I have to pointed that additional tests of the porosity will strongly increase the manuscript quality. If it is not possible, please clearly state why did not you make this examination.
Authors reply:
We appreciate the reviewer’s comment. We test the porosity of samples using the liquid absorption method. The test method is detailed in in ref 33. The original PAN membrane’s porosity is 96.8%. The hydrolyzed PAN-nanofiber membrane (4 hours)’s porosity is 84.5%. The nanofiber membrane after embedding PDA (20 hours)’s porosity is 81.31%.
Comment 2:
Table with emulsion content description is necessary. Additional details should be presented nor in the materials and methods section.
Authors reply:
We thank the reviewer for providing constructive feedbacks to improve our manuscript. We have added the tables in the revised manuscript to description the comparison.
Comment 3:
Did you use some surfactants/stabilizers for emulsion preparation?
Authors reply:
We thank the reviewer for the useful comments. This is indeed a very important research point. Regrettably, this article did not make further research on this aspect. We plan to focus on this aspect in further research.
Comment 5:
Please explain what role plays here polydopamine?
Authors reply:
We thank the reviewer for the useful comment. The effect of PDA nanocluster is discussed in terms of improved wettability due to hydroxyl groups. We did choose to use PDA based on such considerations of all adhesion and simple preparation method, and added relevant expositions in the “introduction”, and cited articles in related fields to support our choice.
Comment 6:
Additional tests which will show the oil content (or other components) before and after filtration are necessary to prove the membrane efficiency.
Authors reply:
We appreciate the reviewer’s comment. The filtration efficiency of this article is indeed calculated by testing the TOC content before and after filtration. The corresponding filtration efficiency and TOC content after filtration have been given in section 2.3 of this article. Although it is described in “Materials and Methods” section that the emulsions were prepared by mixing oils and water under vigorous stirring at a speed of approximately 2000 rpm, a mass ratio of 1:99. However, this 1wt% content data is not used as the data of the content of emulsion before filtration, but the TOC content of each emulsion is tested separately as the TOC content of the emulsion before filtration.
Comment 7:
Abstract - please avoid the words such as "outstanding, excellent, etc." you can use a high efficiency, etc.
Authors reply:
We thank the reviewer for providing constructive feedbacks to improve our manuscript. We have revised some inappropriate wording in the manuscript.
Comment 8:
It must be pointed, that I really enjoyed all figures - most of them are well organized. I m truly happy, that the Authors showed how the system works, as presented in fig. 5. Nevertheless, please add the values on the scale bars as well as error bars. Please verify fig. 5d where is before and what is after permeation?
Authors reply:
We thank the reviewer for providing constructive feedbacks to improve our manuscript. We have reorganized the order of the figures in the revised manuscript. Arranged the figures in order of discussion. Figure 5 was modified. The values on the scale bars is mentioned in the figure’s annotation.
Comment 9:
Fig. 3f please enlarge the Ra images.
Authors reply:
We appreciate the reviewer’s suggestion. We have reorganized the order of the figures in the revised manuscript. Arranged the figures in order of discussion. And the Ra images was enlarged in the revised manuscript.
Comment 10:
Uniform the colors on the graphs for each sample.
Authors reply:
We thank the reviewer for providing constructive feedbacks to improve our manuscript. Figure 4(b) was modified in the revised manuscript to be consistent with others.

Round 2
Reviewer 1 Report
In my opinion the authors have improved the manuscript answering to the comments. I just have a few suggestions as minor revisions.
- Line 251, I think that concentration of tris-hydrochloride buffer should be correct in 1 mM.
- I strongly suggest to carefully check OVERALL the text, because I still see a lot of space and punctuation mistakes, even in the new added sentences. Only for some examples: space before square brackets of reference: Line 39, 46, 38, 75 and so on… or points in references: Line 184, etc; uppercase character after point: line 237 (in table description), 273, etc.
Author Response
Response to Editor and Reviewers
Molecules
Manuscript ID: molecules-1201771
Manuscript Title: Polydopamine nanocluster embedded nanofibrous membrane via blow spinning for separation of oil/water emulsions
Dear Editor and Reviewers,
We are grateful for your kind help and constructive suggestions on our manuscript. We have studied the reviewer’s comments carefully and have made the corrections accordingly.
Starting from page 2 in this file, we provide the point-by-point replies to the reviewer’s comments. Meanwhile, the modified contents have been highlighted in the revised manuscript (in red) for your consideration. We hope our efforts on modification will make our work more appropriate for publication.
Thank you for your attention and consideration. We are looking forward to hearing from you.
Sincerely yours,
Minghao Fang
Response to the reviewer 1:
Authors reply:
We thank the reviewer for providing constructive feedback to improve our manuscript. We have gone through the reviewers’ comments carefully and have made the modifications accordingly. We highlighted the modified contents in red in the revised manuscript. Our point-by-point responses to the reviewer’s comments are provided here.
Comment 1:
Line 251, I think that concentration of tris-hydrochloride buffer should be correct in 1 mM.
Authors reply:
We are grateful for the reviewer’s comment. We have checked the supplier’s product information and it is indeed 1M.
Comment 2:
I strongly suggest to carefully check OVERALL the text, because I still see a lot of space and punctuation mistakes, even in the new added sentences. Only for some examples: space before square brackets of reference: Line 39, 46, 38, 75 and so on… or points in references: Line 184, etc; uppercase character after point: line 237 (in table description), 273, etc.
Authors reply:
We appreciate the reviewer’s suggestion. We are sorry for the mistakes in the manuscript. We have carefully examined the paper and revised them.
Reviewer 2 Report
The Authors improved the manuscript due to the Reviewers comments.
In my opinion, the conclusion should be more highlighted the impact and role of polydopamine on the presented system. 1 or 2 sentences will strongly increase the manuscript quality.
Please consider using full names of the components in the conclusions where are using or the first time (in the conclusion section). It might be helpful for readers. This is an only suggestion, not mandatory.
I m recommending this manuscript after just minor additional changes in the conclusions.
Author Response
Response to Editor and Reviewers
Molecules
Manuscript ID: molecules-1201771
Manuscript Title: Polydopamine nanocluster embedded nanofibrous membrane via blow spinning for separation of oil/water emulsions
Dear Editor and Reviewers,
We are grateful for your kind help and constructive suggestions on our manuscript. We have studied the reviewer’s comments carefully and have made the corrections accordingly.
Starting from page 2 in this file, we provide the point-by-point replies to the reviewer’s comments. Meanwhile, the modified contents have been highlighted in the revised manuscript (in red) for your consideration. We hope our efforts on modification will make our work more appropriate for publication.
Thank you for your attention and consideration. We are looking forward to hearing from you.
Sincerely yours,
Minghao Fang
Response to the reviewer 2:
The Authors presented an interesting idea of the membrane composition. First, is this manuscript belong to some special issue? If not, I will consider switching this manuscript for Processes, Applied Sciences, Materials, Membranes, or Nanomaterials. Second, there are a few details that should be explained before publication, and major revision is needed.
Authors reply:
We thank the reviewer for the useful comments. Based on the suggestions, we have revised our work carefully with corresponding corrections and additions. We highlighted the modified contents in the revised manuscript.
Comment 1&3:
In my opinion, the conclusion should be more highlighted the impact and role of polydopamine on the presented system. 1 or 2 sentences will strongly increase the manuscript quality.
I ’m recommending this manuscript after just minor additional changes in the conclusions.
Authors reply:
We thank the reviewer for the useful comment. We have emphasized the enhanced hydrophilicity of the Polydopamine (PDA)-nanocluster on the nanofiber membrane in Conclusions and marked in the manuscript (in red).
Comment 2:
Please consider using full names of the components in the conclusions where are using or the first time (in the conclusion section). It might be helpful for readers. This is an only suggestion, not mandatory.
Authors reply:
Thank you for your kind suggestions. The full names of the SBS and PDA are added in the related location.